# From Investigating a Case of Cellulitis to Exploring Nosocomial Infection Control of ST1 *Legionella pneumophila* Using Genomic Approaches

**DOI:** 10.3390/microorganisms12050857

**Published:** 2024-04-25

**Authors:** Charlotte Michel, Fedoua Echahidi, Sammy Place, Lorenzo Filippin, Vincent Colombie, Nicolas Yin, Delphine Martiny, Olivier Vandenberg, Denis Piérard, Marie Hallin

**Affiliations:** 1Department of Microbiology, Universitair Ziekenhuis Brussel (UZ Brussel), Vrije Universiteit Brussel (VUB), Laarbeeklaan 101, 1090 Brussels, Belgium; 2Department of Microbiology, Laboratoire Hospitalier Universitaire de Bruxelles (LHUB-ULB), Rue Haute 322, 1000 Brussels, Belgium; 3Department of Internal Medicine and Infectious Diseases, EpiCURA Hospital, 7301 Hornu, Belgium; 4Faculty of Medicine and Pharmacy, Mons University, Chemin du Champ de Mars 37, 7000 Mons, Belgium; 5Innovation and Business Development Unit, Laboratoire Hospitalier Universitaire de Bruxelles (LHUB-ULB), Rue Haute 322, 1000 Brussels, Belgium; 6Centre for Environmental Health and Occupational Health, School of Public Health, Université Libre de Bruxelles (ULB), Avenue Roosevelt 50, 1050 Brussels, Belgium; 7European Plotkin Institute for Vaccinology (EPIV), Université Libre de Bruxelles (ULB), Avenue Roosevelt 50, 1050 Brussels, Belgium

**Keywords:** *Legionella pneumophila*, *Legionella pneumophila* ST1, whole genome sequencing, genomic typing, wgMLST, cgMLST, wgSNP, nosocomial, hygiene investigation

## Abstract

*Legionella pneumophila* can cause a large panel of symptoms besides the classic pneumonia presentation. Here we present a case of fatal nosocomial cellulitis in an immunocompromised patient followed, a year later, by a second case of Legionnaires’ disease in the same ward. While the first case was easily assumed as nosocomial based on the date of symptom onset, the second case required clear typing results to be assigned either as nosocomial and related to the same environmental source as the first case, or community acquired. To untangle this specific question, we applied core-genome multilocus typing (MLST), whole-genome single nucleotide polymorphism and whole-genome MLST methods to a collection of 36 Belgian and 41 international sequence-type 1 (ST1) isolates using both thresholds recommended in the literature and tailored threshold based on local epidemiological data. Based on the thresholds applied to cluster isolates together, the three methods gave different results and no firm conclusion about the nosocomial setting of the second case could been drawn. Our data highlight that despite promising results in the study of outbreaks and for large-scale epidemiological investigations, next-generation sequencing typing methods applied to ST1 outbreak investigation still need standardization regarding both wet-lab protocols and bioinformatics. A deeper evaluation of the *L. pneumophila* evolutionary clock is also required to increase our understanding of genomic differences between isolates sampled during a clinical infection and in the environment.

## 1. Case Reports/Background

*Legionella pneumophila* (*L. pneumophila*) is a ubiquitous waterborne pathogen mostly known to cause mild to severe pneumonia with a mortality rate around 10% [1]. It can also be involved in extra-pulmonary infections such as cardiological or neurological, hematopoietic disorders, hepatitis or cellulitis [2,3]. The bacteria infect patients through water reservoirs and air-cooling systems after being inhaled. Diverse species and subtypes of *Legionella* colonize water and form biofilms in pipelines, but *L. pneumophila* sequence type (ST) 1 and 47 are the main types found in clinical cases of Legionnaire’s disease (LD) in Europe [4,5]. *Legionella* remain a threat to public health because even if infection episodes are mostly community-acquired (CA) and sporadic, clusters and epidemics often happen, despite a tight control of *L. pneumophila* ratios in water systems [5,6]. In Belgium, each case must be declared to public health authorities and leads to an environmental investigation by public health institutions to identify the source of contamination and prevent epidemics.

The increasing incidence of LD in Europe is of concern, and the current reference typing technique for comparing *L. pneumophila* isolates (sequence-based typing, SBT) is not discriminatory enough to always guarantee an epidemiological link between isolates, mostly when applied to ST-1. A recent nosocomial case of cellulitis caused by *L. pneumophila* ST-1 led us to evaluate the capacities of existing typing methods for ST-1 isolates.

### 1.1. Case 1

A patient (P1) was hospitalized for deterioration of general status, in a secondary hospital (H1) in Wallonia, Belgium, in October 2019. The patient was also treated for metastatic cancer for two years, recently complicated by an episode of febrile neutropenia. On day 16, the patient developed signs of sepsis, and large-spectrum antibiotics (glycopeptides and third-generation cephalosporin) were initiated. Within 24 h, P1 was admitted to the intensive care unit as he became hypoxemic and developed spontaneous and rapidly progressing abdominal wall cellulitis and signs of systemic shock. An X-ray performed at that time showed bilateral condensing infiltrates with right pleurisy. In this context, a urinary *L. pneumophila* antigen test was performed and turned out to be positive. Interestingly, this was also the case for a liquid sample collected from a small third space close to the cellulitis. Moxifloxacin was immediately added to the treatment, but the patient died a few hours later.

Four clinical samples were sent to the Belgian National Reference Center (NRC) for culture and PCR detection of *L. pneumophila*: endotracheal aspirate (ETA), pleural fluid, ascites fluid and a cutaneous biopsy sample. All were positive for *L. pneumophila* both by PCR and culture. The isolates were all serogroup 1 (Sg1) and sequence-based typing (SBT) sequence-type (ST) 1.

As the patient had no recent travel history, no activity suggesting contamination via a recreational water system and was hospitalized for more than two weeks before the onset of symptoms, a diagnostic of nosocomial infection was retained.

The patient did not have the physical capacity to leave his bedroom, so the water distribution system of the room was sampled as the principal suspected source of contamination. Culture of the sink’s hot and cold-water system samples reached 50,000 colony-forming units (CFU) and 20,000 CFU/L of *L. pneumophila* Sg1, respectively, and confirmed to be ST1 by the NRC. The water system was then decontaminated by heat shock following protocols, and control samples were negative [7].

### 1.2. Case 2

One year later, another immunocompromised patient (P2) was hospitalized in the same ward. On day 10 after hospitalization, they were diagnosed with clinical pneumonia caused by *L. pneumophila*, which was ST1 according to SBT (broncho-alveolar lavage isolate). The patient was successfully treated and left the hospital shortly after. As 10 days is the usual breakpoint used for nosocomial onset, an environmental investigation was initiated. P2 did not stay in the same room that P1 had occupied, and two isolates of *L. pneumophila* ST1 were cultured from the water systems of their bedroom. The water system was decontaminated, and control samples were negative. While an epidemiological link between environmental isolates and clinical isolates from P2 was difficult to clarify here, the additional question of a link between the two clinical cases was triggered as the persistence of a strain in the hospital environment is a well described possibility [8,9].

At present, when an investigation is set up, SBT is the reference technique for comparing *L. pneumophila* isolates. However, as illustrated by the two cases described above, when the isolates investigated belong to ST1, the epidemiological link can rarely be confirmed [1]. In Belgium, 19% of *L. pneumophila* infections are related to Sg1 ST1, one of the five major STs found worldwide [4,5,6,10]. Unlike the other four major lineages of *L. pneumophila* (ST23, ST37, ST47 and ST62), ST1 is ubiquitously found in the environment and very often isolated from environmental investigations in water systems following both CA and nosocomial *L. pneumophila* cases [4,10]. In addition, *Legionella* forms biofilms and lives at a slow pace until factors favoring its multiplication occur [8,11]. We thus decided to investigate the correlation between the isolates from both episodes by core-genome (cg) multilocus sequence typing (MLST), to extend the number of alleles included in the comparison, using the pattern designed by Moran-Gilad et al. [12] that serves as a reference for many European studies on *Legionella* (Table 1).

Three out of four clinical samples from P1, the two environmental matching isolates and the isolates found in the environmental investigations of P2 clustered together when applying a maximum four allelic differences (ADs) between two isolates [12] (Figure 1). The clinical isolate from P2 is five ADs away from the closest isolates, putting it at the limit of being included in the cluster. Interestingly, the P1 isolate LEG1116 is 6 ADs away from the closest isolate, 9 to 10 ADs from isolates sampled from the same patient on the same day and 11 ADs away from the clinical isolate of P2. Consequently, along with the practical question of a possible link between these two specific cases through the persistence of a strain in the ward environment [8], the question of the adequate technique and the adequate threshold to infer reliable conclusions regarding ST1 isolates of *L. pneumophila* relationship was raised.

Since next-generation sequencing (NGS) has been developed, its use as a high-resolution typing tool to distinguish closely related isolates has contributed to improve infection control and outbreak management for a wide range of bacterial pathogens [1,9,12,13,14,15,16]. Applied to *L. pneumophila*, cg and whole-genome (wg) single nucleotide polymorphism (SNP) and wg and cg-MLST have been reported as showing good discrimination performances [1,9,12,17,18]. Nevertheless, issues remain for highly represented STs like ST1 [8,12,15,16]. Indeed, ST1 is known to be both genetically highly conserved and ubiquitous, but its population structure remains poorly described [19]. We carried out a mini review of *L. pneumophila* genomic studies conducted since 2015 in order to explore the methods and clustering thresholds used to investigate both outbreaks and long-term surveillances of *L. pneumophila* to identify which to apply to our specific question (Table 1).

As the pattern for cgMLST with the proposed threshold of four ADs used as reference in most studies did not give conclusive results using the nine isolates described above (Figure 1), a larger analysis was performed (Figure 2). We decided to test several methods described in the literature such as cgMLST, wgSNP and wgMLST on a panel comprising the nine previous isolates and 27 well documented Belgian ST1 isolates collected between 1985 and 2020. These isolates were collected during nosocomial ST1 epidemics that took place in the 1980s in two other Belgian hospitals: Hospital 2 (H2) (six isolates, of which two were clinical) and Hospital 3 (H3) (13 isolates; 11 clinical) and from a fourth hospital (H4: two clinically related isolates). Five unrelated (one nosocomial, four CA) and one environmental isolate from H3 were also used. To complete the panel, we included 41 ST1 isolates from nine countries collected between 1992 and 2018 (Appendix A).

#### 1.2.1. cgMLST of a Panel of 77 ST1 Isolates

Most isolates grouped together by country (Figure 2A). Regarding Belgian isolates, three groups appeared (Figure 3). The well-identified epidemic isolates from H2 and H3 were at maximum separated by five ADs except for the H3 isolate LEG515, (>50 ADs from the H3 closest isolate) which is opposed to the conclusion made by the initial investigation but consistent with the monoclonal subtyping performed at the time (Philadelphia versus Benidorm) [20]. However, in the same study, LEG517, also subtyped Philadelphia, belonged to the H3 cluster [20] (Appendix A). To cluster together isolates known to be epidemiologically linked (H3 epidemic isolates, H2 epidemic isolates, and P1 isolates), a threshold of six ADs should be applied (Figure 3). Then, all nine clinical and environmental isolates from H1 would also cluster together. However, the H1 cluster would also involve LEG767 (an epidemiologically unrelated 2017 CA sporadic case from the same region), the H3 LEG515 isolate, and the two H4 isolates (located in the same city as H3) from 15 years earlier (2003–2004). Similarly, the H3 epidemic cluster would include an environmental isolate from the same hospital (LEG723), but from 30 years later (2016—Appendix A) and a sporadic CA isolate acquired in the same city as H3 (LEG325).
microorganisms-12-00857-t001_Table 1Table 1Mini review of the literature on *Legionella pneumophila* genomic studies beginning in 2015.Analysis SettingReferenceYearNumber of Isolates ComparedST1 Included?Cluster-Defining Retained Ratio (Maximum SNP or AD)Number of Alleles ComparedPurpose of StudycgMLSTMoran-Gilad et al. [12]201515Yes4 AD1521NA (*)Petzold et al. [17] 201745No4 AD ($)1521OutbreakWüthrich et al. [1]201994No10 AD1521OutbreakGorzynski et al. [9]20223397Yes115 AD1469EpidemiologyRicci et al. [18] 202363No4 AD ($)1346 (+)OutbreakwgSNPLapierre et al. [21]201777Yes4 SNP ($)NAOutbreakDavid et al. [8]2017229Yes4 SNPNAOutbreakSchjørring et al. [22]201712No4 SNPNAOutbreakWells et al. [23]201828Yes4 SNPNAEpidemiologyRaphael et al. [19]2019113Yesno ratioNAEpidemiologycgSNPBartley et al. [24]201646Yesno ratioNot mentionedOutbreakQin et al. [25]201653Yesno ratio1896EpidemiologyBuultjens et al. [26]2017180No9 SNPNAOutbreakRaphael et al. [19]201930Yes4 SNP ($)Not mentionedOutbreakGorzynski et al. [9]20223397Yes16 SNP1469EpidemiologyRicci et al. [18]202363No4 SNP ($)Not mentionedOutbreakwgMLSTRaphael et al. [27]201630Yes>98% similarity Not mentionedOutbreakRaphael et al. [19]2019113Yes>98% similarity 5778Epidemiology(*): serves as technical reference for cgMLST, ($): observed by writer, (+): adapted from the cgMLST reference, AD: allelic difference, SNP: single nucleotide polymorphism, MLST: multilocus sequence typing, cg: core-genome, wg: whole-genome.


#### 1.2.2. wgSNP Analysis of the Panel of 77 ST1 Isolates

A wgSNP analysis was then run on the same group of strains (Figure 2B). The commonly proposed SNP threshold in the literature is four SNPs. The minimum spanning tree (Figure 4) shows that this threshold does not cluster together either the H3 or the P1 isolates. The fitting threshold would be eight SNPs, which would allow more discrimination than cgMLST. Indeed, while still showing clusters by country of origin on the large scale (Figure 2B), this threshold allows for H2 and H3 epidemics to form well-defined clusters, excluding LEG723 from the H3 cluster but not LEG 325. In this analysis, the P2 clinical isolate stays outside the cluster formed by all other H1 isolates with a difference of 11 SNPs, as opposed to the H4 isolates, H3 LEG515 and the unrelated sporadic LEG767 isolate, which remain inside the cluster (Figure 4).

#### 1.2.3. wgMLST Analysis of the ST1 Isolates

The last analysis was the wgMLST pattern from the Bionumerics software (Figure 2C and Figure 5). Of note, only an average of 52% of the supposed 5770 alleles were called (Appendix A). Lacking references to settle a threshold (Table 1), a threshold of 12 ADs was set up based on the maximum distance between known linked H3, H2 and P1 isolates (Figure 5). All isolates from the H2 and H3 epidemic then cluster together, while the two sporadic LEG723 and LEG 325 cluster with the H3 epidemic. The P2 isolate, 18 loci away from its closest relative, was clearly distant from the H1 cluster, which, by contrast, included (as with wgSNP) sporadic LEG767 and H4 isolates.

## 2. Discussion

The incidence of *L. pneumophila* infections has been steadily increasing since the 2000’s [5,6]. This phenomenon, suspected to be directly or indirectly linked to climate change [28], is expected to continue increasing in the coming years, which stresses the necessity of finding efficient typing methods for environmental investigations and prevention of epidemics. *L. pneumophila* has a wide genomic diversity and is easily subject to recombination events, mostly within a same serogroup [29,30]. *L. pneumophila* is also able to spread over several distinct geographical areas without acquiring substantial genomic diversity over time [1,9,16,30]. The hypothesis of intermittent replication of *L. pneumophila* was mentioned after several large epidemiological investigations spanning diverse countries and several decades [9,23]: the cycle of life of *L. pneumophila* involves a latent phase that would explain both this intermittence phenomenon and the low rate of genomic polymorphism over space and time [9,23,26,31].

All STs of *L. pneumophila* seem to have evolved separately over time, and ST1, one of the best fitted lineages for infections and adaptation in the environmental niches created by humans, is one of the two STs that show the slowest genomic evolution [1,32]. This results in the phenomenon that even highly discriminatory typing methods can cluster together isolates known to be epidemiologically unrelated.

Here we have reported two cases of LD caused by ST1 isolates in patients hospitalized in the same ward one year apart. The first was easily proven as nosocomial based on the date of symptom onset, whereas the other case required clear typing results to be either assigned as nosocomial and related to the same environmental source as the first one, or to CA. To untangle this specific question, we applied several previously described NGS-based typing methods and thresholds to a collection of ST1 isolates retrieved either from documented nosocomial epidemics or sporadic infections in Belgium and ST1 genomes published onto repository online databases. The commonly used cgMLST with a maximum four AD threshold [12] was somewhat too stringent, as several isolates of the H3 outbreak were separated by five ADs. According to David et al. [33], ST1 and other major STs are better discriminated by the SNP approach, using a threshold of four SNPs, than by the cgMLST, although it is slightly variable according to the ST studied (Table 1). In our setting, this threshold was also too stringent, as isolates proved and published [19] to be linked in space and time (H2 and H3 outbreaks, P1 clinical isolates) can be separated by up to eight SNPs from their closest relative (Figure 2B). This underlines the importance, mentioned by others, of environmental sampling and the usefulness of space and time epidemiological data for validating the interpretation of WGS results for LD investigations [9].

Regardless of the method used, unrelated isolates, acquired in the same geographical areas but years apart, did cluster together as witnessed for sporadic LEG723 and H3 isolates, or isolates from H4 and LEG767, which are from the same city. These isolates could be linked as a possibility of a long-lasting, intermittently replicating reservoir in a common water system is considered. For instance, David et al. demonstrated that cases acquired in a community close to a hospital can show the same genotype as nosocomial cases in this hospital [8]. Indeed, the water reservoirs of a city can be contaminated by the same predominant clone of *L. pneumophila*, and thus distribute it in several geographically close plumbing networks [11]. Similarly, an Australian study demonstrated that outbreak clones had a maximum difference of ten SNPs within a common cg-SNP ST30 clade in a 30 km radius area around Melbourne [26]. The relatedness of *L. pneumophila* types and subtypes in both water reservoirs of health-care facilities and their respective municipal water systems should be regularly investigated by quantitative methods and NGS typing in order to better understand and evaluate the *L. pneumophila* ecology within large water distribution systems over long periods of time.

The unusual availability of more than one sequenced isolate for P1 gives an unexpected perspective to our analysis. These isolates (collected the same day, from the same patient but from different body sites) differed by up to 12 SNPs (wgSNP) and 13 ADs (cgMLST).

This difference observed between same-day P1 isolates, as compared with H1 environmental isolates that are one year apart, could reflect the differential evolutionary clock of the ST1 strain when infecting a patient as compared with dormancy in a pipe. Usually, when an investigation by NGS methods is set up, one isolate per patient is compared to one isolate from a suspected source. If the diversity observed among P1 isolates does usually occur in patients, this might induce a bias in analysis performed using a tight discrimination scale. On the other hand, most SNPs found between *L. pneumophila* isolates are >90% related to homologous recombination events that occur in a very limited area of the genome [9,31]: the genes encoding activation factors of the Type 4 Secretion System. This was the case in the nine P1 and P2 related isolates. When compared all together, H1 isolates (clinical and environmental) are separated by 354 SNPS with the strict SNP filtering (inter-SNP distance of min 12 bp, absolute coverage of 5 minimum, removal of non-informative SNPs, ambiguous bases, unreliable bases and gaps), which goes up to 537 SNPs when the inter-SNP distance and non-informative SNPs filters are removed. This observation supports that the shift between environment to clinical setting may not be related to punctual scattered mutations but rather localized rearrangements linked to the fitness of the bacteria. In such cases, a large number of SNPs would appear in a single genetic event, thus making SNP filtering key to the performance of a wgSNP.

The necessity of standardized quality metrics for NGS typing methods also deserves comments. Raw data found in online repositories can be used by different users with different approaches. We excluded several genomic sequences from our analysis after bad quality results for de novo assemblies based on N50 < 100,000, which was generally associated with discordant genome length and/or high N bases numbers. Nevertheless, the very same sequences were used by others that considered them to be of reliable quality after using another de novo alignment protocol (Appendix A) [9,34]. In datasets with such slow genetic variation and for which a single SNP can be so meaningful, data quality should be more under focus. The inference of epidemiological links due to unreliable quality of sequence data is of concern, as it may generate legal implications for epidemics and nosocomial cases. For SBT, an online tool is available to everyone and is used by all European NRCs in order to type their isolates and is always given with a quality score. Such a tool should be developed to help WGS to be standardized for public health applications.

Using epidemiologically linked H3 and P1 isolates, wgSNP and cgMLST analyses gave contradictory results regarding the link of P2 isolate with P1 and H1 environmental isolates. wgMLST is described as a good tool for large epidemiological investigations, but less if applied to ST1 [19]. As described by David et al. [33], by taking more genes into account, wgMLST indexes more dissimilarities between isolates than cgMLST, but also needs a much better quality of sequencing to be sure of the allelic call. Furthermore, as the number of alleles differs between studies, the results obtained are not comparable. Here, even limited to an average of 52% of supposedly 5770 alleles called, the results matched wgSNP, in agreement with the observations of David et al. A tailored threshold based on known epidemiological links between isolates resulted in good discrimination both on a large scale and in the clustering of well described Belgian epidemics apart from sporadic isolates (Figure 5) and suggested that the P2 isolate is not linked to P1 and H1 environmental isolates.

## 3. Conclusions

Our data highlight that despite promising results in the study of outbreaks and for large-scale epidemiological investigations, NGS typing methods applied to ST1 *L. pneumophila* outbreak investigation still need standardization regarding both wet-lab protocols and bioinformatics. Indeed, for several important varia like reference strain, quality metrics of *de novo* alignment, filtering of SNPs and threshold for clustering, no reference exists. As the incidence of *L. pneumophila* infections is increasing steadily, there is a need for methods that allow reliable discrimination of ST1 isolates. To this end, a deeper evaluation to increase our understanding of the *L. pneumophila* evolutionary clock during both active clinical infection and persistence in the environment is also required.

## 4. Materials and Methods

### 4.1. Isolate Preparation, Routine Diagnostics and Typing

In Belgium, all case of Legionnaires’ disease (LD) must be reported to the federal health institute. An environmental investigation is then undertaken to identify the source. Belgian laboratories can send, on a voluntary basis, both samples and *Legionella* isolates to the National Reference Centre for identifying *Legionella pneumophila* either for diagnostic by PCR or SB typing. Cultures are performed on BCYE with GVPC agar in a humid atmosphere at 35 °C +/− and incubated for 48–72 h. Sequence-based typing is performed according to the European Legionnaires’ Disease Surveillance Network’s (ELDSNet) method (http://bioinformatics.phe.org.uk/legionella/legionella_sbt/php/sbt_homepage.php, accessed on 16 April 2024) [35].

### 4.2. Whole-Genome Sequencing

Extraction of total DNA was performed using the Qiagen DNeasy blood & tissue kit (Qiagen, Hilden, Germany) from growth colonies of each isolate. DNA concentration was assessed using a Qubit dsDNA HS (or BR) assay kit (ThermoFisher Scientific, Waltham, MA, USA) and a Qubit 2.0 Fluorometer (ThermoFisher Scientific). Library preparation was done with the Kapa HyperPlus Library Preparation Kit (Kapa Biosystems, Wilmington, MA, USA). Quality control and pooling of the library was performed employing a 2100 BioAnalyzer (Agilent, Santa Clara, CA, USA), the Qubit 2.0 Fluorometer (ThermoFisher Scientific) and the KAPA Illumina Library Quantification Kit (Kapa Biosystems). The library was pooled with the addition of a 1% PhiX control library after denaturation with 0.2 N NaOH, to a final concentration of 2 nM. MiSeq or Hiseq (Illumina, San Diego, CA, USA) sequencers were used with the MiSeq Reagent kit v2 (500 cycle; 2 × 250 bp read-length) the expected coverage was of 100 [36].

### 4.3. Bioinformatic Analysis

The raw data were uploaded as fastq files on the software Bionumerics v8.1 (Biomérieux©, Marcy-l’Étoile, France). De novo assembly was done with a SPAdes algorithm; for one isolate (LEG1116), as quality metrics were not achieved by SPAdes, assembly by another scheme also based on a DeBruijn graph was used: SKESA [37,38]. The reference genome was downloaded from the NCBI website: *L. pneumophila* Paris 1 (Accession number (AN): CR628336). Quality scores of each method for each isolate are provided in Appendix A. N-50 was superior to 100,000, Q-score at least 30 (32–38), and average de novo covering was superior to 40 (46–203) [9,34].

wgSNP was performed by remapping the reads to a reference genome, and then a strict SNP filtering was applied (inter-SNP distance of minimum 12 bp, absolute coverage of 5 minimum, removal of non-informative SNPs, ambiguous bases, unreliable bases and gaps). wgMLST was performed by both free-based calls and assembly-based calls with the plugin of the software with default settings and screened 5770 alleles of the genome. cgMLST was calculated based on the previous analysis and covered 1521 loci based on the definition by Moran-Gilad et al. (10). The number of alleles called for both wgMLST and cgMLST was calculated with the statistical plugin of the BioNumerics software.

Phylogenic analyses were made on the BN software by MST for categorical data analysis. Branches were logarithmically scaled according to the distance between each node.

All genomes of Belgian isolates are publicly available on the NCBI website in the project PRJNA1073851. Five Fastq files for isolates from other countries were downloaded from the European Nucleotide Archive from projects listed in Appendix A, and four isolates were excluded because of quality scores.

## Figures and Tables

**Figure 1 microorganisms-12-00857-f001:**
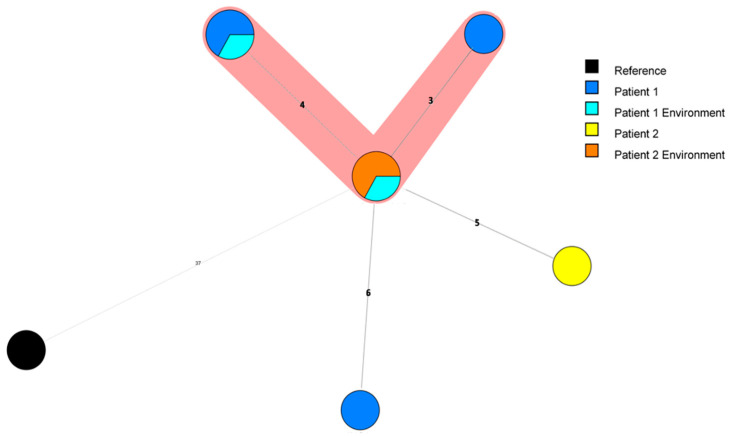
Minimum spanning tree for categorical data of the cgMLST analysis based on the call of 1521 alleles [12] performed using Bionumerics software v8.1 on nine isolates from Hospital 1, including environmental isolates related to Patients 1 and 2. A cluster analysis of maximum 4 allelic differences between two isolates is highlighted in red. Branch lengths are logarithmically scaled and allelic distances are written on branches.

**Figure 2 microorganisms-12-00857-f002:**
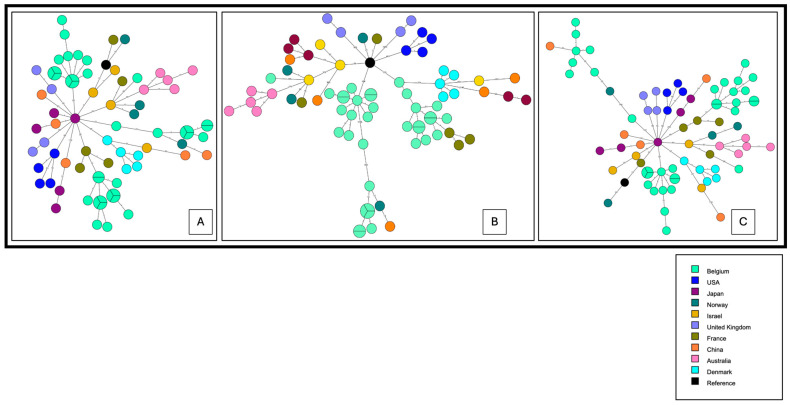
Minimum spanning tree performed on Bionumerics software v8.1. ST1 isolates of *L. pneumophila (n* = 77) are represented by nodes colored according to their country of origin. (**A**): cgMLST. (**B**): wgSNP with strict SNP filtering. (**C**): wgMLST (5770 alleles). Branch lengths are logarithmically scaled and allelic distances are written on branches.

**Figure 3 microorganisms-12-00857-f003:**
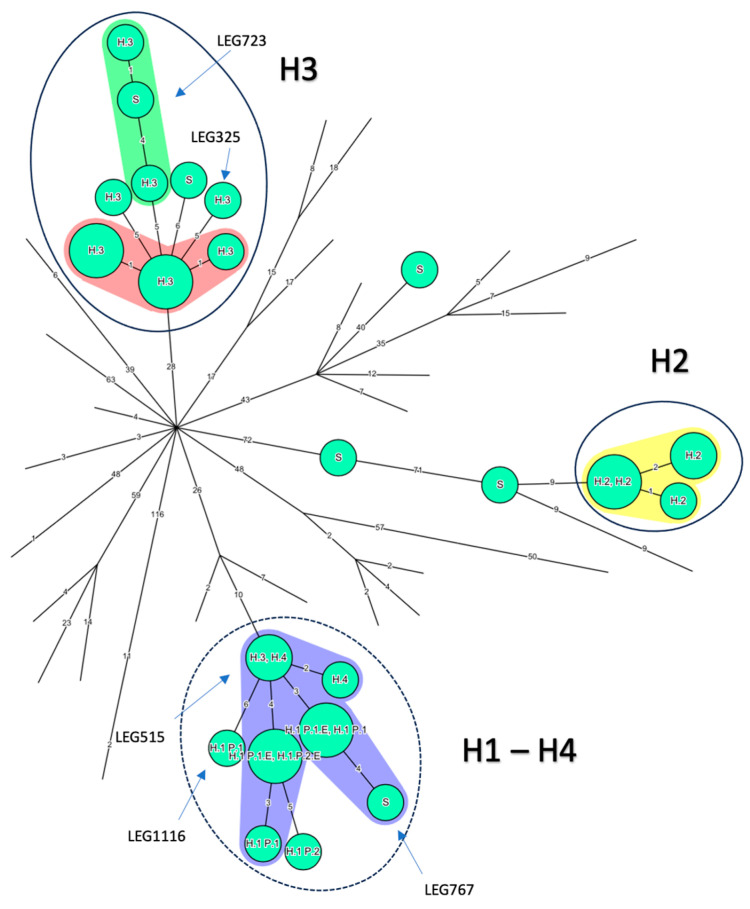
Minimum spanning tree for categorical data of the cgMLST analysis based on the call of 1521 alleles [12] performed using Bionumerics software v8.1 on a panel of 77 ST1 isolates. *L. pneumophila* Paris 1 was used as a reference. Nodes representing non-Belgian isolates were reduced to uncover the relationship between the 36 Belgian ST1 isolates. The setting of each isolate is written above each node: Hospital 1,2, 3 and 4: H1, H2, H3 and H4, respectively; sporadic cases: S; Patient 1 and 2: P1 and P2, respectively; environmentally linked isolates: E. Clustering performed using a distance of four allelic differences (ADs) between isolates appears against a colored background. The solid blue line corresponds to the threshold of six ADs proposed by authors to include well characterized epidemics in H2 and H3. The cluster obtained for H1 isolates using these six ADs is circled in a blue dashed line, and includes one H3 isolate and the two H4 isolates. Branch lengths are logarithmically scaled and allelic distances are written on branches.

**Figure 4 microorganisms-12-00857-f004:**
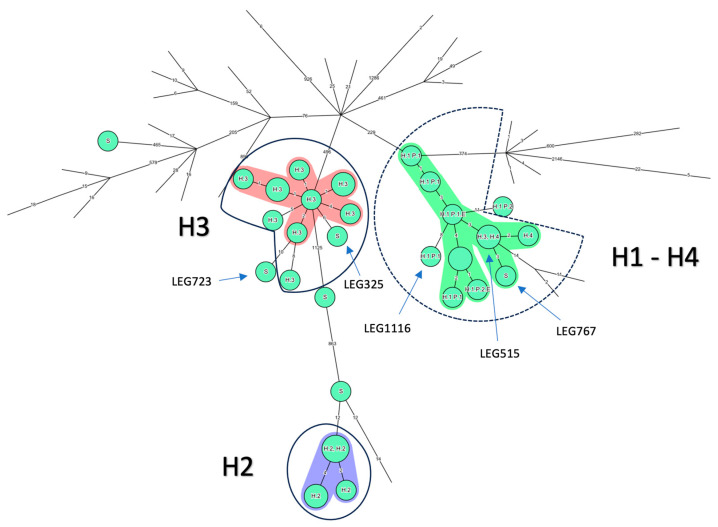
Minimum spanning tree of wgSNP with strict SNP filtering performed by mapping the reference strain *L. pneumophila* Paris 1 on Bionumerics software v8.1. Nodes representing non-Belgian isolates were reduced to uncover the relationship between the 36 Belgian ST1 isolates. The setting of each isolate is written above each node: Hospital 1,2, 3 and 4: H1, H2, H3 and H4, respectively; sporadic cases: S; Patient 1 and 2: P1 and P2, respectively; environmentally linked isolates: E. Clustering performed using a four allelic differences (ADs) distance between isolates appears against a colored background. The solid blue line corresponds to the threshold of eight SNPs proposed to include well characterized epidemics in H2 and H3. The cluster obtained for H1 isolates using these eight SNPs is circled in a blue dashed line, entails one H3 isolate and the two H4 isolates, but excludes the P2 clinical isolate. Branch lengths are logarithmically scaled and allelic distances are written on branches.

**Figure 5 microorganisms-12-00857-f005:**
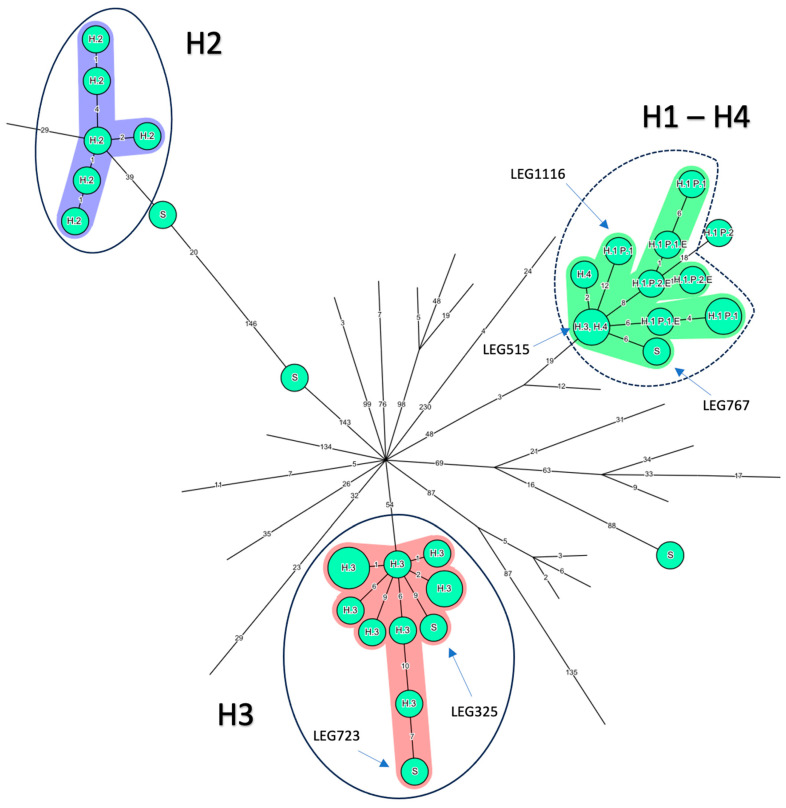
Minimum spanning tree of categorical data analysis for wgMLST performed on Bionumerics software v8.1. Nodes representing non-Belgian isolates were reduced to undercover only the relationship between the 36 Belgian ST1. The setting of each isolate is written above each node. Hospital 1,2, 3 and 4: H1, H2, H3 and H4, respectively; sporadic cases: S; Patient 1 and 2: P1 and P2, respectively; environmentally linked isolates: E. Clustering was performed using a distance of 12 allelic differences between isolates and appears against a colored background within the groups. Solid blue line shows a well-defined cluster formed for H2 and H3. Dashed blue line shows that the suspected cluster excludes the P2 clinical isolate with the chosen threshold. Branch lengths are logarithmically scaled and allelic distances are written on branches.

## Data Availability

Michel, Charlotte (Forthcoming 2024). From investigating a case of cellulitis to exploring nosocomial infection control of ST1 *Legionella pneumophila* using genomic approaches. [Dataset]. Dryad. https://doi.org/10.5061/dryad.7m0cfxq35.

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
