# Peer review of "From Investigating a Case of Cellulitis to Exploring Nosocomial Infection Control of ST1 Legionella pneumophila Using Genomic Approaches"

_microorganisms, 2024, doi:10.3390/microorganisms12050857_

Round 1
Reviewer 1 Report
Comments and Suggestions for Authors
Authors need to complete the minor issues before acceptance of the manuscript.
1- The key word is Legionella pneumophila. Check for italics.
2- As the author mentions in line 72, the contamination of water: what are the possible causes or sources of contamination?
3-UFC/L: is it the same as colony-forming units (CFU) or different? If different, then you need to explain it in line 74.
4- As the author mentioned, ST1 is ubiquitously found in the environment and very often isolated from environmental sources—air, water, or soil.?
5-Line no. 291 As described by David et al. (27), need to check.
6- Conclusions section can be improved more significantly; also mention why NGS typing methods applied to the ST1 outbreak investigation still need standardization. What kind of standardization needs to be focused on for better use in the future.
Comments on the Quality of English LanguageOk
Author Response
Dear reviewer, thank you for your thorough comments and suggestions. We have modified the manuscript in response to your feedback. Please find below point-by-point our responses to your comments:
- The key word is Legionella pneumophila. Check for italics.
We adapted the keywords and reviewed the whole document accordingly.
- As the author mentions in line 72, the contamination of water: what are the possible causes or sources of contamination?
As mentioned in the first paragraph, Legionella are waterborne pathogen, so the question of community-acquired infection via any water source or air-cooling systems was addressed. As the patient was hospitalized for 14 days, in his bedroom, without even using the shower because of health status, only the water system from the sink of his own bedroom was suspected.
The sentence “The patient was not in the physical capacity of leaving his bedroom so the water distribution system of the room was sampled as principal suspected source of contamination” was added
- UFC/L: is it the same as colony-forming units (CFU) or different? If different, then you need to explain it in line 74.
Indeed our abbreviation was badly written and was corrected as recommended
- As the author mentioned, ST1 is ubiquitously found in the environment and very often isolated from environmental sources—air, water, or soil.?
To enhance the understanding of this topic, the first paragraphs of the introduction were modified. See L46-47
5-Line no. 291 As described by David et al. (27), need to check.
The citation was adapted.
- Conclusions section can be improved more significantly; also mention why NGS typing methods applied to the ST1 outbreak investigation still need standardization. What kind of standardization needs to be focused on for better use in the future.
We modified the conclusion according to your request, see L273-279, L323-330 and L345-350.
Reviewer 2 Report
Comments and Suggestions for Authors
Adjust the type of citations to the journal.
In general, Table 1 and the figures have low resolution and lack description.
Check the use of acronyms, Lpne and L. pneumophila
Adjust the writing to the magazine format
-The authors should be more descriptive in their clinical cases, sociodemographic variables, follow-up, and patient outcomes
-The authors should expand the discussion on how genetic variability affects the pathogenicity and transmission of L. pneumophila.
-Delve into the study limitations section, explicitly addressing the implications of variations in sequencing quality and how this could affect the interpretation of the results.
Author Response
Dear reviewer, thank you for your thorough comments and suggestions. We have modified the manuscript in response to your feedback. Please find below point-by-point our responses to your comments:
- Adjust the type of citations to the journal.
The adjustment has been made
- In general, Table 1 and the figures have low resolution and lack description.
All figures were re-formated with a much improved definition. Figure 1 was divided and a new organization of the figures is proposed. We hope the quality of them will meet your expectations
- Check the use of acronyms, Lpne and L. pneumophila
Thank you for noticing this mistake, all the abbreviations were changed to L. pneumophila
- Adjust the writing to the magazine format
The citations have been put into square brackets.
- The authors should be more descriptive in their clinical cases, sociodemographic variables, follow-up, and patient outcomes
We agree that a case report should as detailed as possible, we added clinical details for patient 2 and environmental investigations in order to be clearer in our description. However, we have voluntarily hidden details that might affect the anonymity of the patients such as gender, age and localization. First, the local ethical committee was insisting on the most complete achievable anonymity of patients, secondly, we believe that these details do not affect the problems and conclusions we encountered here.
- The authors should expand the discussion on how genetic variability affects the pathogenicity and transmission of L. pneumophila.
Additional description was added. See Lines 300- 306
- Delve into the study limitations section, explicitly addressing the implications of variations in sequencing quality and how this could affect the interpretation of the results.
Additional description was added. See Lines 317- 326
Reviewer 3 Report
Comments and Suggestions for Authors
Legionella pneumophila can cause a wide range of symptoms beyond classical pneumonia, and a comprehensive study of this microorganism is of particular practical and fundamental interest. I have a couple of comments on this article.
1. The entire article lacks a bit of scientific writing. Especially, the relevance of the topic and the assigned tasks are written in a very general way. And the discussion part should also be expanded.
2. The quality of all illustrations and tables is very poor. Please improve the quality.
Author Response
Dear reviewer, thank you for your thorough comments and suggestions. We have modified the manuscript in response to your feedback. Please find below point-by-point our responses to your comments:
Legionella pneumophila can cause a wide range of symptoms beyond classical pneumonia, and a comprehensive study of this microorganism is of particular practical and fundamental interest. I have a couple of comments on this article.
- The entire article lacks a bit of scientific writing. Especially, the relevance of the topic and the assigned tasks are written in a very general way. And the discussion part should also be expanded.
To enhance the understanding of the relevance of this topic, first paragraphs of the introduction were modified. Additional description was added between Lines 300- 306. Additional description was added between Lines 317- 326
- The quality of all illustrations and tables is very poor. Please improve the quality.
All figures were re-formated with a much improved definition. Figure 1 was divided and a new organization of the figures is proposed. We hope the quality of them will meet your expectations
Reviewer 4 Report
Comments and Suggestions for Authors
I was invited to review the article entitled “From investigating a case of cellulitis to exploring nosocomial infection control of ST1 Legionella pneumophila using genomic approaches.” In my opinion this paper has a hybrid format, tackling two different issues: a case series of two patients who developed Legionella pneumophila infection and an epidemiologic analysis of those isolates who is focusing however on the issues related to molecular typing of the ST1 Legionella pneumophila. I believe that this paper should be split in two, a case report/case series of the two patients and a review/original article tackling the epidemiologic issues of typing ST1 Legionella pneumophila. However, if the authors want to proceed with this structure, I have some suggestions that I will present as follows:
· Perhaps there is some editing error with Figure 1 but at the current form it is very hard to follow since it seems split in 3. There seems to be no beginning and no end to it. I would kindly recommend addressing these issues and perhaps split this figure in 3 instead of presenting that organisation in A-B-C.
· Table 1 is just a picture added. It has a poor resolution, so the text is not accessible. I cannot assess the relevance of the table, or the data in the current format. Please address this.
· Line 236 - Why did you mentioned that it was assumed that the infections was nosocomial? Wasn’t that already proven?
· Line 221- You used the abbreviation Lpm. What does it stand for?
· Line 247 – You mentioned that the technique was too stringent because you knew that some of the isolated were linked. By what method were you able to assess that those strains are related? Are they really related since the tools that you described here do not seem to support this. Please explain.
Although this paper tackles an interesting topic, I do not think it is fit for publication at the current form. However, I strongly encourage resubmission after careful evaluation and better structure of the text. Best of luck.
Author Response
Dear reviewer, thank you for your comments and suggestions. We tried to answer them as follow:
I was invited to review the article entitled “From investigating a case of cellulitis to exploring nosocomial infection control of ST1 Legionella pneumophila using genomic approaches.” In my opinion this paper has a hybrid format, tackling two different issues: a case series of two patients who developed Legionella pneumophila infection and an epidemiologic analysis of those isolates who is focusing however on the issues related to molecular typing of the ST1 Legionella pneumophila. I believe that this paper should be split in two, a case report/case series of the two patients and a review/original article tackling the epidemiologic issues of typing ST1 Legionella pneumophila. However, if the authors want to proceed with this structure, I have some suggestions that I will present as follows:
- Perhaps there is some editing error with Figure 1 but at the current form it is very hard to follow since it seems split in 3. There seems to be no beginning and no end to it. I would kindly recommend addressing these issues and perhaps split this figure in 3 instead of presenting that organisation in A-B-C.
All figures were re-formated with a much improved definition. Figure 1 was divided and a new organization of the figures is proposed. We hope the quality of them will meet your expectations
- Table 1 is just a picture added. It has a poor resolution, so the text is not accessible. I cannot assess the relevance of the table, or the data in the current format. Please address this.
Table 1 was reshaped to be more suited for lecture.
- Line 236 - Why did you mentioned that it was assumed that the infections was nosocomial? Wasn’t that already proven?
The word assumed was modified to “proven” for a better understanding. (now line 271)
- Line 221- You used the abbreviation Lpm. What does it stand for?
Thank you for noticing this mistake, all abbreviations were changed to L. pneumophila
- Line 247 – You mentioned that the technique was too stringent because you knew that some of the isolated were linked. By what method were you able to assess that those strains are related? Are they really related since the tools that you described here do not seem to support this. Please explain
We modified the text to “In our setting, this threshold was also too stringent, as isolates proved and published [19] to be linked in space and time (H2 and H3 outbreaks, P1 clinical isolates) can be separated by up to 8 SNPs from their closest relative (Figure 2B). This underlines the importance, mentioned by others, of environmental sampling and the usefulness of space and time epidemiological data for validating the interpretation of WGS results for LDs investigations [15].”
Although this paper tackles an interesting topic, I do not think it is fit for publication at the current form. However, I strongly encourage resubmission after careful evaluation and better structure of the text. Best of luck.
We sincerely acknowledge your reserve on the format of this paper. We found ourselves in a pickle between relating our routine-based experience for translation of WGS data to day-to-day evaluation and writing an original scientific paper. This is the reason why we chose such an hybrid format to relate these.
Round 2
Reviewer 2 Report
Comments and Suggestions for Authors
The authors have resolved my comments. The article can be published.
Reviewer 4 Report
Comments and Suggestions for Authors
The quality of the presentations improved considerable. The authors were kind enough to explain, edit and consider all my concerns. The tables and figures were edited and better explained. Altough the format of this paper is uncanny, I would accept for publication considering the lack of existing information in the subject and the acute need for extensive research in the future.